# Distinct expression requirements and rescue strategies for *BEST1* loss- and gain-of-function mutations

Qingqing Zhao[1,2†], Yang Kong[3†], Alec Kittredge[4], Yao Li[3], Yin Shen[5*], Yu Zhang[3*], Stephen H Tsang[6*], Tingting Yang[2,3*]

[1]Eye Center, Renmin Hospital of Wuhan University, Wuhan, China; [2]Department of Pharmacology and Physiology, University of Rochester, School of Medicine and Dentistry, Rochester, United States; [3]Department of Ophthalmology, Vagelos College of Physicians & Surgeons, Columbia University, New York, United States; [4]Department of Pharmacology, Columbia University, New York, United States; [5]Eye Center, Medical Research Institute, Renmin Hospital, Wuhan University, Wuhan, China; [6]Jonas Children's Vision Care, Departments of Ophthalmology and Pathology & Cell Biology, Edward S. Harkness Eye Institute, Institute of Human Nutrition and Columbia Stem Cell Initiative, New York Presbyterian Hospital/ Columbia University Irving Medical Center, New York, United States

**Abstract** Genetic mutation of the human *BEST1* gene, which encodes a $Ca^{2+}$-activated $Cl^-$ channel (BEST1) predominantly expressed in retinal pigment epithelium (RPE), causes a spectrum of retinal degenerative disorders commonly known as bestrophinopathies. Previously, we showed that BEST1 plays an indispensable role in generating $Ca^{2+}$-dependent $Cl^-$ currents in human RPE cells, and the deficiency of BEST1 function in patient-derived RPE is rescuable by gene augmentation (Li et al., 2017). Here, we report that *BEST1* patient-derived loss-of-function and gain-of-function mutations require different mutant to wild-type (WT) molecule ratios for phenotypic manifestation, underlying their distinct epigenetic requirements in bestrophinopathy development, and suggesting that some of the previously classified autosomal dominant mutations actually behave in a dominant-negative manner. Importantly, the strong dominant effect of *BEST1* gain-of-function mutations prohibits the restoration of BEST1-dependent $Cl^-$ currents in RPE cells by gene augmentation, in contrast to the efficient rescue of loss-of-function mutations via the same approach. Moreover, we demonstrate that gain-of-function mutations are rescuable by a combination of gene augmentation with CRISPR/Cas9-mediated knockdown of endogenous *BEST1* expression, providing a universal treatment strategy for all bestrophinopathy patients regardless of their mutation types.

**\*For correspondence:**
yinshen@whu.edu.cn (YS);
yz3802@cumc.columbia.edu (YZ);
sht2@cumc.columbia.edu (SHT);
ty2190@cumc.columbia.edu (TY)

†These authors contributed equally to this work

## Introduction

Bestrophinopathies are a group of five retinal degeneration disorders caused by genetic mutations in the human *BEST1* gene, namely Best vitelliform macular dystrophy (BVMD) (*Marquardt et al., 1998*; *Petrukhin et al., 1998*), autosomal recessive bestrophinopathy (ARB) (*Burgess et al., 2008*), adult-onset vitelliform dystrophy (AVMD) (*Allikmets et al., 1999*; *Krämer et al., 2000*), autosomal dominant vitreoretinochoroidopathy (ADVIRC) (*Yardley et al., 2004*), and retinitis pigmentosa (RP) (*Davidson et al., 2009*). Clinical phenotypes of bestrophinopathies include serous retinal detachment, lesions that resemble egg yolk, or vitelliform, and progressive vision loss that can potentially lead to blindness (*Johnson et al., 2017*). To date, over 250 distinct *BEST1* mutations have been

identified from bestrophinopathy patients, but their pathological mechanisms remain unclear. The majority of the *BEST1* mutations are autosomal dominant, whereas the autosomal recessive ones are specifically linked to ARB (*Johnson et al., 2017*). As there is no effective treatment for bestrophino-pathies yet, dissecting the molecular bases of different *BEST1* mutations is critical for rational design of therapeutic strategies (*Yang et al., 2015*).

Functionally, bestrophin-1 (BEST1), the protein encoded by *BEST1*, is a $Ca^{2+}$-activated $Cl^-$ channel (CaCC) predominantly expressed in retinal pigment epithelium (RPE) (*Marmorstein et al., 2000*). Bestrophinopathy patient-derived RPE cells exhibit abnormal $Ca^{2+}$-dependent $Cl^-$ currents, under-scoring the indispensable role of BEST1 as a CaCC in RPE (*Li et al., 2017*), although the contribution of other candidate CaCCs cannot be excluded. Structurally, while the human BEST1 structure has not been solved, high-resolution structures of three homologs from *Klebsiella pneumoniae* (KpBEST), chicken (cBEST1), and bovine (bBEST2) indicate that the channel is a highly conserved pentamer with a flower vase-shaped ion conducting pathway (*Yang et al., 2014a*; *Kane Dickson et al., 2014*; *Owji et al., 2020*).

A key question regarding the pathology of bestrophinopathies is how each *BEST1* mutation spe-cifically affects the channel activity, eventually resulting in retinal degeneration. The vast majority of the tested patient-derived mutations exhibited a loss-of-function phenotype, as the $Cl^-$ currents mediated by the mutant channels are significantly reduced compared to the wild-type (WT) BEST1 (*Li et al., 2017*; *Hartzell et al., 2008*; *Johnson et al., 2014*; *Ji et al., 2019a*; *Ji et al., 2019b*). We recently identified several gain-of-function mutations, which enhance the channel activ-ity when transiently expressed in HEK293 cells but still cause bestrophinopathy (*Ji et al., 2019a*), suggesting the physiological importance of maintaining normal BEST1 functionality. However, although most loss-of-function and all gain-of-function mutations known so far are autosomal domi-nant, it remains elusive whether they have different capacities to influence the channel activity in the presence of WT BEST1, as one would expect in heterozygous carriers. In general, gain-of-function mutations often display a stronger dominant effect than loss-of-function mutations, but a side-by-side comparison between them has not been conducted for *BEST1*. This is essential for evaluating the pathogenicity of different *BEST1* mutations, especially considering that allelic expression imbalance (AEI) at the *BEST1* locus has been observed in human RPE (*Llavona et al., 2017*). More-over, the strength of the mutations' dominant effect is critical for gene augmentation therapy, as higher augmentation dosages may be necessary to suppress stronger mutations.

In this study, we quantitatively examined the functional influence of different classes of patient-derived mutations on the channel when the mutant and WT BEST1 were co-expressed at various ratios in HEK293 cells. Strikingly, all six autosomal dominant loss-of-function mutations behaved recessively at a 1:1 ratio with the WT BEST1 and required a superior 4:1 ratio to exhibit the mutant phenotype. It suggests that they act in a dominant-negative manner rather than the canonical domi-nant manner, which explains our previous results that gene augmentation is sufficient for the rescue of autosomal dominant loss-of-function mutations (*Ji et al., 2019b*). Consistent with this finding, the mutant *BEST1* allele is transcribed at a higher level than the WT allele in patient-derived RPE cells. In sharp contrast, all three autosomal dominant gain-of-function mutations displayed a dominant behavior, even at an inferior 1:4 ratio with the WT BEST1. Due to their strong dominant effect, *BEST1* gain-of-function mutations cannot be rescued by gene augmentation alone, but require CRISPR/Cas9-mediated silencing of the endogenous *BEST1* in combination with gene augmentation for restoring $Ca^{2+}$-dependent $Cl^-$ currents in RPE cells. Additionally, we confirmed the physiological role of BEST1 as the *bona fide* CaCC in RPE. Taken together, our results reveal the differences between loss- and gain-of-function mutations, and provide a therapeutic strategy for all *BEST1* mutations.

## Results

### *BEST1* loss-of-function mutations affect Cl⁻ currents in a dosage-sensitive manner

To quantitatively evaluate the influence of *BEST1* mutations on the channel activity under a condition mimicking the endogenous gene dosage, seven YFP-tagged BEST1 loss-of-function mutants, including six autosomal dominant (A10T, R218H, L234P, A243T, Q293K, and D302A) and one autosomal recessive (P274R), were individually mixed with CFP-tagged WT BEST1 at a 1:1 ratio and introduced into HEK293 cells for patch clamp recording. Surprisingly, in the presence of 1.2 µM free intracellular $Ca^{2+}$ ($[Ca^{2+}]_i$), where BEST1 conducts peak current (*Li et al., 2017*), Cl⁻ currents from cells co-expressing mutant and WT BEST1 were similar to those from cells expressing WT BEST1 alone (*Figure 1a–h* cyan, *Figure 1—figure supplement 1a*, *Figure 1—figure supplement 2*), regardless of whether the mutation is autosomal dominant or recessive. Therefore, these six loss-of-function mutations, although genetically defined as autosomal dominant, do not exhibit dominant behavior *in vitro*.

To test if a dominant-negative effect is at play, the mutants were individually co-transfected with WT BEST1 at a 4:1 ratio into HEK293 cells for patch clamp analysis. At 1.2 µM $[Ca^{2+}]_i$, Cl⁻ currents from co-expression of an autosomal dominant mutant and WT BEST1 were significantly smaller than those from the WT only, and similar to those from the mutant only (*Figure 1b–g* magenta, *Figure 1—figure supplement 1b*). By contrast, currents from cells co-expressing the autosomal recessive P274R mutant and the WT BEST1 at a 4:1 ratio were still similar to those from cells expressing the WT BEST1 only (*Figure 1h*, *Figure 1—figure supplement 1b*). Therefore, the six previously

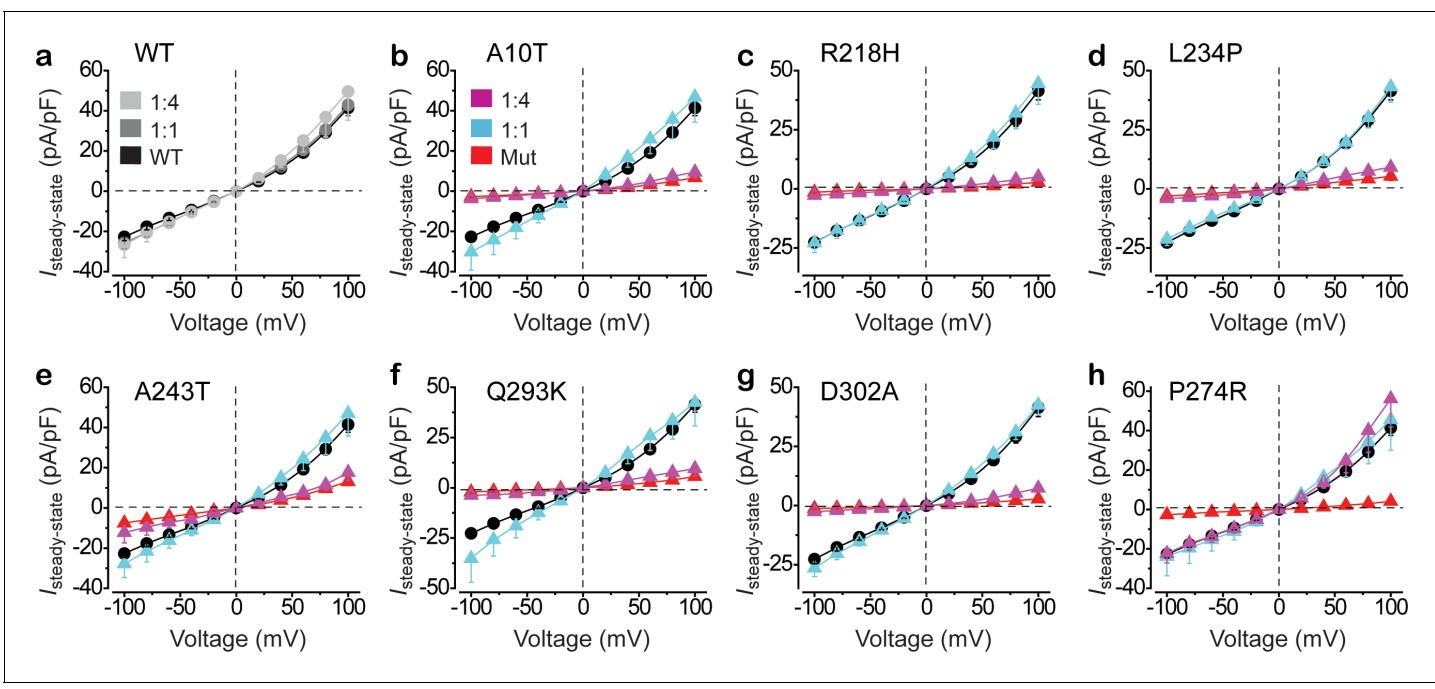

**Figure 1.** Functional influence of BEST1 loss-of-function mutants in HEK293 cells. (**a**) Population steady-state current density-voltage relationships in HEK293 cells expressing BEST1 WT-CFP only (black), WT-CFP: WT-YFP = 1:1 (gray), or WT-CFP: WT-YFP = 1:4 (light gray), in the presence of 1.2 µM $[Ca^{2+}]_i$, n = 5–6 for each point. (**b–h**) Population steady-state current density-voltage relationships in HEK293 cells expressing BEST1 WT-CFP: mutant-YFP = 1:1 (cyan), WT-CFP: mutant-YFP = 1:4 (magenta), compared to mutant (red) or WT (black) only, in the presence of 1.2 µM $[Ca^{2+}]_i$, n = 5–6 for each point. The mutants are BEST1 A10T (**b**), R218H (**c**), L234P (**d**), A243T (**e**), Q293K (**f**), D302A (**g**), and P274R (**h**). All error bars in this figure represent s.e.m. See also *Figure 1—figure supplements 1* and *2*.

The online version of this article includes the following figure supplement(s) for figure 1:

**Figure supplement 1.** Electrophysiological analysis of BEST1 loss-of-function mutations.

**Figure supplement 2.** Patient-derived *BEST1* mutations in a homology model.

recognized autosomal dominant mutations are actually dominant-negative *in vitro*, whereas the autosomal recessive P274R mutation indeed behaves recessively.

## Imbalanced transcription of *BEST1* alleles in human RPE

Our patch clamp results from transiently transfected HEK293 cells predict that the autosomal dominant mutant allele is expressed at a higher level than the WT allele in patients' RPE, such that the dominant-negative effect can be manifested. To test this hypothesis, we extracted mRNA from patient-derived induced pluripotent stem cell (iPSC) differentiated RPE (iPSC-RPE), and measured the ratio of transcripts from the mutant and WT *BEST1* alleles by reverse transcription polymerase chain reaction (RT-PCR) and TOPO cloning. Remarkably, the mutant genotype showed up three to four times more than the WT in all 12 BVMD patient-derived iPSC-RPE clones (two clones from each patient) (*Table 1*), indicating that the transcription level of the mutant allele is three- to fourfold higher than that of the WT allele in these patients' RPE cells.

To further validate if the two *BEST1* alleles have imbalanced transcription in native RPE, we collected RPE cells from a post-mortem donor harboring heterozygosity of a single nucleotide polymorphism (SNP, rs767552540) in *BEST1*. Consistent with results from iPSC-RPE, transcripts from one allele outnumbered those from the other by approximately threefold in these native human RPE cells (*Table 1*).

Together, our results suggest that allelic imbalance of *BEST1* transcription contributes to the dominant-negative effect of the autosomal dominant mutations. Importantly, this provides an explanation for the restoration of $Ca^{2+}$-dependent $Cl^-$ currents by gene augmentation in iPSC-RPE cells bearing a *BEST1* autosomal dominant loss-of-function mutation (*Ji et al., 2019b*): as long as the augmented WT BEST1 protein is expressed at a similar or higher level compared to the endogenous BEST1, the mutant to WT protein ratio is no longer in a dominant-negative scenario, such that the WT phenotype is exhibited as seen in 1:1 transiently transfected HEK293 cells (*Figure 1b–g*).

## *BEST1* gain-of-function mutations are *bona fide* dominant *in vitro*

We previously identified three BEST1 gain-of-function mutations, namely D203A, I205T, and Y236C, all of which are autosomal dominant (*Figure 1—figure supplement 2*; *Ji et al., 2019a*). To test whether their behavior is dominant *in vitro*, each mutant was individually co-expressed with WT at 1:1 in HEK293 cells and subjected to patch clamp analysis. Without $Ca^{2+}$, $Cl^-$ currents from cells co-expressing WT BEST1 and any of the mutants were significantly larger than those from cells expressing WT BEST1 only; at 1.2 μM $[Ca^{2+}]_i$, cells co-expressing D203A/WT and Y236C/WT displayed significantly bigger currents than WT only (*Figure 2a–c left*, *Figure 2—figure supplement 1a*); at both conditions, currents from cells co-expressing mutant/WT BEST1 resembled those from cells expressing the mutant only (*Figure 2a–c*, *Figure 2—figure supplement 1a*). These results indicate that these three gain-of-function mutations are indeed dominant, in contrast to the dominant-negative behavior of the six loss-of-function mutations.

**Table 1.** Sequencing of BEST1 transcripts in retinal pigment epithelium (RPE) cells.
#1–6 are patient-derived iPSC-RPE cells carrying the same set of *BEST1* mutations as those analyzed in transiently transfected HEK293 cells in *Figure 1*. #7 is native human RPE cells from a healthy donor bearing a single nucleotide polymorphism (SNP) in the *BEST1* gene.

| Donor # | Mutation | RPE type | Mutant/WT from clone #1 | Mutant/WT from clone #2 |
|---------|----------|----------|-------------------------|-------------------------|
| 1 | A10T | iPSC-RPE | 72/23 | 51/12 |
| 2 | R218H | iPSC-RPE | 84/20 | 45/11 |
| 3 | L234P | iPSC-RPE | 77/19 | 42/20 |
| 4 | A243T | iPSC-RPE | 83/28 | 37/11 |
| 5 | Q293K | iPSC-RPE | 76/19 | 46/10 |
| 6 | D302A | iPSC-RPE | 78/18 | 35/14 |
| 7 | rs767552540 | Native | 74/23 | NA |

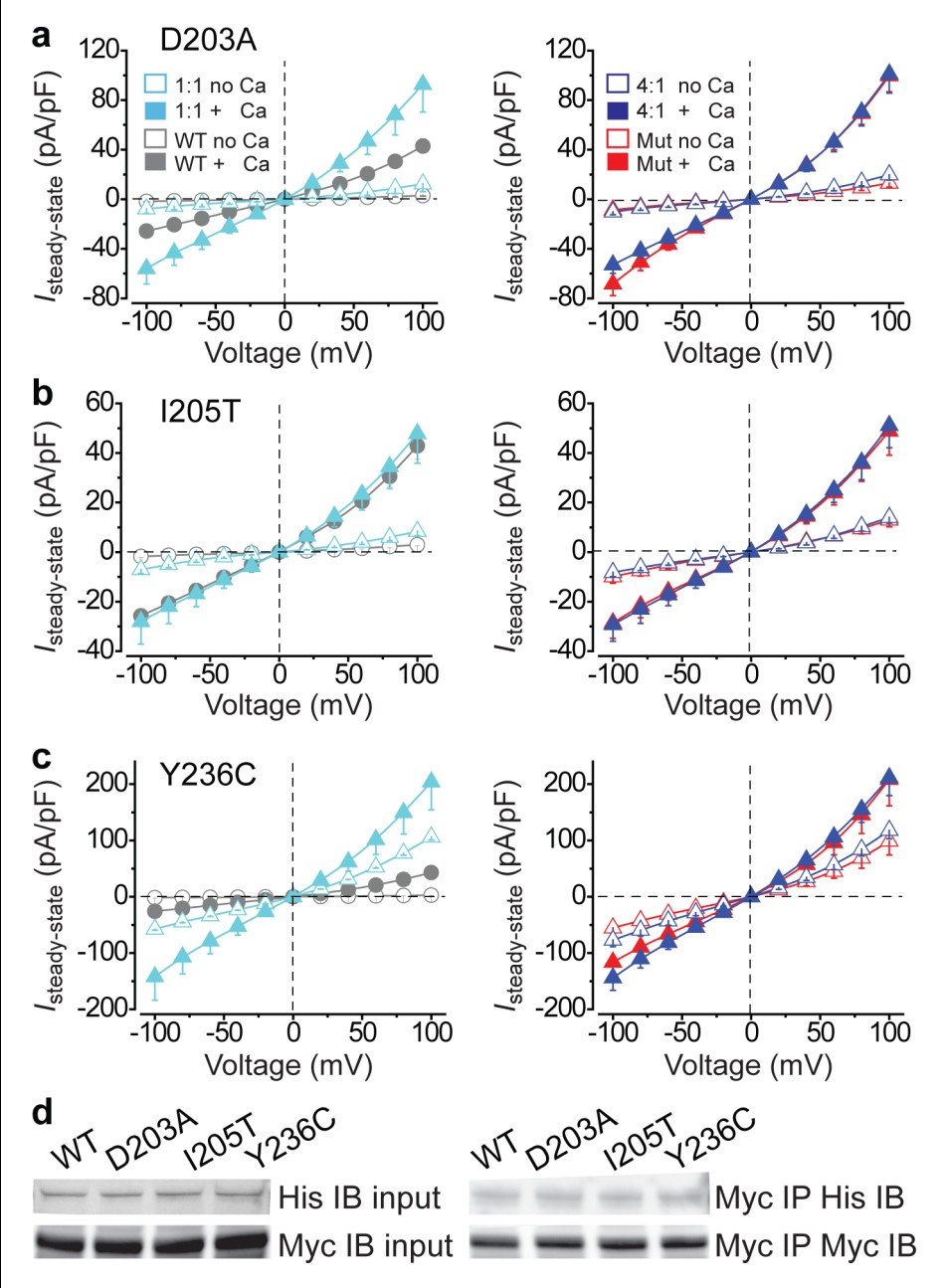

**Figure 2.** Functional influence of BEST1 gain-of-function mutants in HEK293 cells. (**a–c**) *Left*, population steady-state current density-voltage relationships in HEK293 cells co-expressing WT-CFP: mutant-YFP = 1:1 (cyan) compared to WT only (WT-CFP: WT-YFP = 1:1, gray), in the absence (open) or presence (solid) of 1.2 μM $[Ca^{2+}]_i$, n = 5–6 for each point. *Right*, population steady-state current density-voltage relationships in HEK293 cells co-expressing WT-CFP: mutant-YFP = 4:1 (blue) compared to mutant only (red), in the absence (open) or presence (solid) of 1.2 μM $[Ca^{2+}]_i$, n = 5–6 for each point. The mutants are BEST1 D203A (**a**), I205T (**b**), and Y236C (**c**). All error bars in this figure represent s.e.m. (**d**) WT or mutant BEST1-YFP-His was co-expressed with WT BEST1-CFP-Myc in HEK293 cells, and detected by immunoblotting directly in cell lysate (input) or after co-immunoprecipitation. See also *Figure 2—figure supplements 1* and *Figure 2—source data 1*.

The online version of this article includes the following source data and figure supplement(s) for figure 2:

**Source data 1.** The uncropped blots in *Figure 2d* and *Figure 3—figure supplement 1*.

**Figure supplement 1.** Electrophysiological analysis of BEST1 gain-of-function mutations.

Since BEST1 is presumably a pentamer based on known bestrophin structures (*Yang et al., 2014a*; *Kane Dickson et al., 2014*; *Owji et al., 2020*), it is possible that as few as one gain-of-function mutant monomer in the pentameric assembly could alter the channel function. To test this idea, HEK293 cells were co-transfected with mutant/WT BEST1 at a 1:4 ratio for patch clamp analysis. Under this condition, $Ca^{2+}$-dependent $Cl^-$ currents from co-expression of a gain-of-function mutant and WT BEST1 were still similar to those from the mutant only (*Figure 2a–c right*, *Figure 2—figure supplement 1b*). These results suggest a potent dominant effect of the gain-of-function mutations: just one mutant monomer is sufficient to dominate the function of the pentameric channel. To confirm the interaction between the gain-of-function mutant and WT monomers, mutant BEST1-YFP-His and WT BEST1-CFP-Myc were co-expressed in HEK293 cells, followed by immunoprecipitation with an antibody against Myc and immunoblotting with antibodies against His and Myc, respectively. All three gain-of-function mutants were expressed at similar levels to that of WT BEST1 after transient transfection, and retained the interaction with WT BEST1 (*Figure 2d*), consistent with our previous observation that the interaction between BEST1 monomers is not affected by loss-of-function autosomal dominant mutations (*Ji et al., 2019b*).

## Modeling *BEST1* gain-of-function mutations in hPSC-RPE cells

We previously showed that WT gene augmentation is sufficient to restore $Ca^{2+}$-dependent $Cl^-$ currents in iPSC-RPE cells with a *BEST1* loss-of-function mutation, while the exogenous BEST1 is expressed at a comparable level to the endogenous protein (*Ji et al., 2019b*). As BEST1 gain-of-function mutations are dominant over the WT even at a 1:4 ratio (*Figure 2a–c right*, *Figure 2—figure supplement 1b*), it raises an important question on the efficacy of gene augmentation. However, iPSC-RPE cells bearing a gain-of-function mutation are currently unavailable due to the lack of patient samples.

To circumvent this obstacle, we generated isogenic RPE cells (human pluripotent stem cell [hPSC] derived RPE [hPSC-RPE]) from an H1 background hPSC line carrying an inducible Cas9 cassette (H1-iCas9), which allows efficient genome editing (*González et al., 2014*; *Moshfegh et al., 2016*; *Idelson et al., 2009*). The RPE status of the hPSC-RPE cells was recognized by morphological signatures including intracellular pigment and hexagonal shape, and confirmed by immunoblotting with RPE-specific marker proteins RPE65 (retinal pigment epithelium-specific 65 kDa protein) and CRALBP (cellular retinaldehyde-binding protein) (*Figure 3—figure supplement 1a*; *Moshfegh et al., 2016*), consistent with the results from donor-derived iPSC-RPE (*Figure 3—figure supplement 1b*). $Ca^{2+}$-dependent $Cl^-$ currents on the plasma membrane of $BEST1^{WT/WT}$ hPSC-RPE cells were recorded as $4 \pm 1$ and $267 \pm 79$ pA/pF at 0 and 1.2 μM $[Ca^{2+}]_i$, respectively, consistent with results from donor-derived $BEST1^{WT/WT}$ iPSC-RPE (*Figure 3a*). To evaluate the genetic dependency of $Ca^{2+}$-dependent $Cl^-$ currents in RPE cells, we individually knocked out *BEST1* and three other CaCCs, namely *TMEM16A*, *TMEM16B*, and *LRRC8A* in the H1-iCas9 cell line, and generated the corresponding knockout hPSC-RPE cells for patch clamp analysis. It should be noted that only the mRNA of BEST1, but not of the other three CaCCs, can be detected in WT PSC-RPE or donor native RPE cells (*Figure 3—figure supplement 2a–b*). Remarkably, $Ca^{2+}$-dependent $Cl^-$ current was completely eliminated in $BEST1^{-/-}$ hPSC-RPE and a patient-derived BEST1 null (IVS1 +5G>A *homo*) iPSC-RPE (*Figure 3b*, *Figure 3—figure supplement 3a–d*; *Fung et al., 2015*), in contrast to the WT-like currents from $TMEM16A^{-/-}$, $TMEM16B^{-/-}$, or $LRRC8A^{-/-}$ hPSC-RPE cells (*Figure 3c–e*, *Figure 3—figure supplement 3d*). Consistently, the protein and mRNA levels of BEST1 were abolished in $BEST1^{-/-}$ hPSC-RPE cells, but not affected in $TMEM16A^{-/-}$, $TMEM16B^{-/-}$, or $LRRC8A^{-/-}$ hPSC-RPE cells (*Figure 3—figure supplements 1a* and *2c*). Moreover, when WT BEST1 was expressed from a baculovirus vector in $BEST1^{-/-}$ hPSC-RPE and the patient-derived BEST1 null iPSC-RPE, $Ca^{2+}$-dependent $Cl^-$ currents were fully rescued (*Figure 3b*, *Figure 3—figure supplement 3c*). Taken together, these results validate hPSC-RPE as a model system to study BEST1 function, and indicate that BEST1, but not TMEM16A, TMEM16B, or LRRC8A, is the CaCC conducting $Ca^{2+}$-dependent $Cl^-$ current in human RPE.

To model gain-of-function mutations, we individually introduced heterozygous I205T and Y236C mutations to the *BEST1* gene in the H1-iCas9 cell line, generating $BEST1^{I205T/WT}$ and $BEST1^{Y236C/WT}$ hPSC cells, which were then differentiated to $BEST1^{I205T/WT}$ and $BEST1^{Y236C/WT}$ hPSC-RPE cells, respectively, for patch clamp analysis (*Figure 3—figure supplement 1a*). Consistent with results from transiently transfected HEK293 cells (*Ji et al., 2019a*), $Cl^-$ currents from $BEST1^{I205T/WT}$ hPSC-

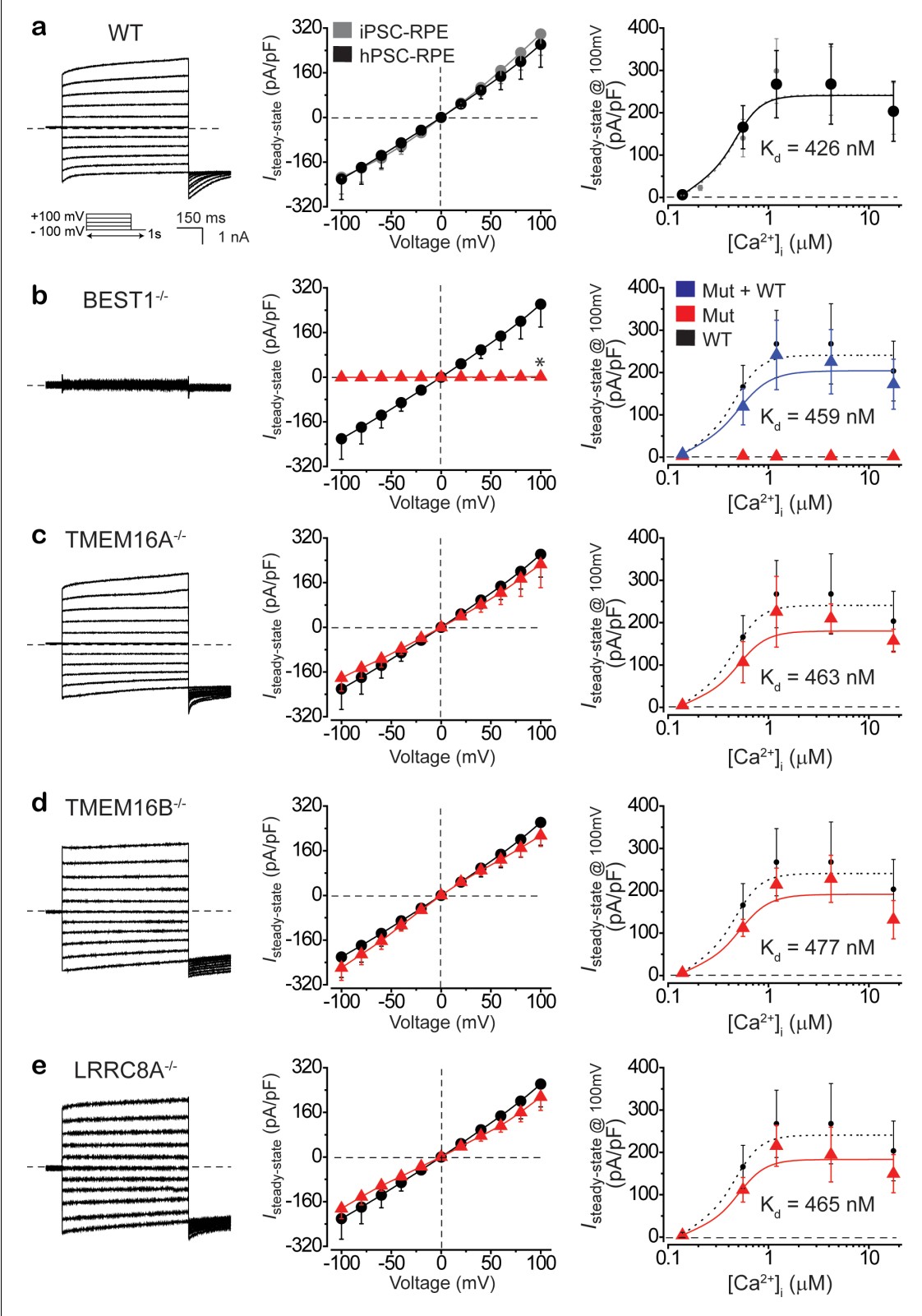

**Figure 3.** BEST1 is responsible for conducting $Ca^{2+}$-dependent $Cl^-$ currents in hPSC-RPE. (**a**) $Ca^{2+}$-dependent $Cl^-$ currents measured by whole-cell patch clamp in WT hPSC-RPE. *Left*, representative current traces recorded at 1.2 μM $[Ca^{2+}]_i$. *Inset*, voltage protocol used to elicit currents. *Middle*, population steady-state current density-voltage relationship in WT hPSC-RPE (black) compared to that from WT iPSC-RPE (gray), at 1.2 μM $[Ca^{2+}]_i$, n = 5–6 for each point. *Right*, steady-state current density recorded at +100 mV plotted vs. $[Ca^{2+}]_i$ from WT hPSC-RPE (black) compared to that from

*Figure 3 continued on next page*

*Figure 3 continued*

WT iPSC-RPE (gray), n = 5–6 for each point. The plot was fitted to the Hill equation. (**b–e**) $Ca^{2+}$-dependent $Cl^-$ currents measured by whole-cell patch clamp in *BEST1$^{-/-}$* (**b**), *TMEM16A$^{-/-}$* (**c**), *TMEM16B$^{-/-}$* (**d**), or *LRRC8A$^{-/-}$* (**e**) hPSC-RPE cells, respectively. *Left*, representative current traces recorded at 1.2 μM $[Ca^{2+}]_i$. *Middle*, population steady-state current density-voltage relationship in knockout hPSC-RPE cells (red), compared to that from WT hPSC-RPE cells (black), at 1.2 μM $[Ca^{2+}]_i$, n = 5–6 for each point. *Right*, steady-state current density recorded at +100 mV plotted vs. $[Ca^{2+}]_i$ from knockout (red) and WT BEST1 supplemented (blue in **b**) hPSC-RPE cells, compared to the plot from WT hPSC-RPE (dotted black), n = 5–6 for each point. Plots were fitted to the Hill equation. *p<0.05 ($1.8 \times 10^{-2}$) compared to WT cells, using two-tailed unpaired Student's *t* test. All error bars in this figure represent s. e.m. See also *Figure 3—figure supplements 1–3* and *Figure 3—source data 1*.

The online version of this article includes the following source data and figure supplement(s) for figure 3:

**Source data 1.** gRNA sequences for CRISPR/Cas9.

**Figure supplement 1.** Expression of RPE-specific marker proteins in hPSC-RPE and iPSC-RPE cells.

**Figure supplement 2.** mRNA levels of $Ca^{2+}$-activated $Cl^-$ channels (CaCCs) in hPSC-RPE cells.

**Figure supplement 3.** $Ca^{2+}$-dependent $Cl^-$ currents in iPSC-RPE and hPSC-RPE cells.

RPE were significantly bigger than those from WT under no or low $Ca^{2+}$ conditions, but similar in the presence of high $Ca^{2+}$ (*Figure 4a–c*, *Figure 3—figure supplement 3d*). On the other hand, the $Ca^{2+}$-dependent $Cl^-$ currents from *BEST1$^{Y236C/WT}$* hPSC-RPE were significantly larger than those from WT at all tested $[Ca^{2+}]_i$s (*Figure 4d–f*, *Figure 3—figure supplement 3d*). These results reaffirm the gain-of-function and dominant behavior of the BEST1 I205T and Y236C mutations in RPE.

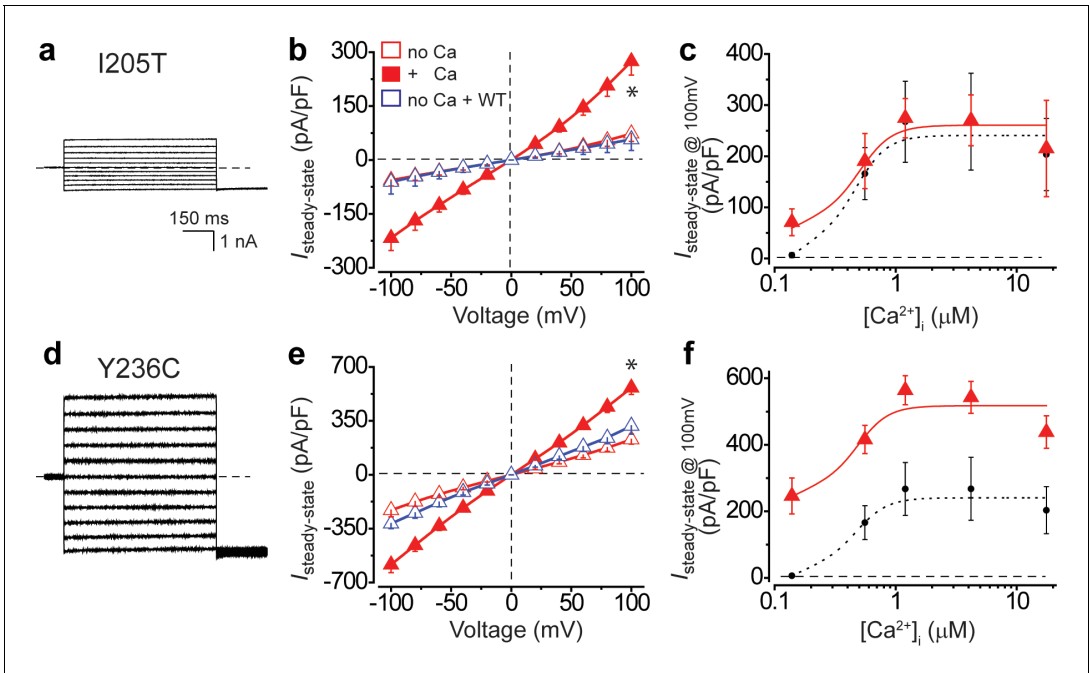

**Figure 4.** $Ca^{2+}$-dependent $Cl^-$ currents in hPSC-RPE cells bearing BEST1 gain-of-function mutations. (**a**) Representative current traces of *BEST1$^{I205T/WT}$* hPSC-RPE in the absence of $Ca^{2+}$. (**b**) Population steady-state current density-voltage relationships in *BEST1$^{I205T/WT}$* hPSC-RPE, in the absence (open red) or presence (solid red) of 1.2 μM $[Ca^{2+}]_i$, compared to cells with WT BEST1 augmentation in the absence of $Ca^{2+}$ (open blue), n = 5–8 for each point. *p<0.05 ($1.3 \times 10^{-3}$) compared to cells without augmentation in the absence of $Ca^{2+}$, using two-tailed unpaired Student's *t* test. (**c**) Steady-state current densities recorded at +100 mV plotted vs. $[Ca^{2+}]_i$ in *BEST1$^{I205T/WT}$* hPSC-RPE (red) compared to those in *BEST1$^{WT/WT}$* hPSC-RPE cells (black), n = 5–6 for each point. (**d–f**) Data for *BEST1$^{Y236C/WT}$* in the same format as (**a–c**), respectively. *p<0.05 ($2.5 \times 10^{-5}$) compared to cells without augmentation in the absence of $Ca^{2+}$, using two-tailed unpaired Student's *t* test. n = 5–10 for each point. All error bars in this figure represent s.e.m. See also *Figure 4—figure supplement 1*.

The online version of this article includes the following figure supplement(s) for figure 4:

**Figure supplement 1.** CRISPR/Cas9-mediated gene silencing in combination with augmentation.

## *BEST1* gain-of-function mutations cannot be rescued by gene augmentation in hPSC-RPE

To test if the aberrant $Ca^{2+}$-dependent $Cl^-$ current in hPSC-RPE bearing a *BEST1* gain-of-function mutation is rescuable by gene augmentation, $BEST1^{I205T/WT}$ and $BEST1^{Y236C/WT}$ hPSC-RPE cells were infected with baculoviruses expressing WT BEST1-GFP and subjected to patch clamp analysis. Notably, $Ca^{2+}$-dependent $Cl^-$ currents in these mutant hPSC-RPE cells remained aberrant after gene augmentation in the absence of $Ca^{2+}$ (*Figure 4b and e*, open blue), despite the exogenous WT BEST1 being expressed at a higher level to that of the endogenous BEST1 (*Figure 4—figure supplement 1a*). This is in sharp contrast to the restoration of $Ca^{2+}$-dependent $Cl^-$ current in $BEST1^{-/-}$ (*Figure 3b*, *Figure 4—figure supplement 1a*) or loss-of-function mutant RPE cells using the same approach (*Ji et al., 2019b*). Therefore, our results suggest that gene augmentation alone is insufficient to rescue *BEST1* gain-of-function mutations.

## Rescue of *BEST1* gain-of-function mutations by non-selective CRISPR/Cas9-mediated gene silencing in combination with augmentation

There are two strategies to overcome the dominant effect of gain-of-function mutations: (1) specific silencing of the mutant allele and (2) non-selective silencing of both endogenous alleles and simultaneously supplying an exogenous WT gene. We reasoned that the latter is a more general approach as one design can be used for various mutations. For the targeted silencing of endogenous *BEST1*, we employed a programmable transcriptional repressor composed of a nuclease-dead Cas9 (dCas9) fused with a bipartite KRAB–MeCP2 repressor domain in the C-terminus (dCas9-KRAB-MeCP2) (*Yeo et al., 2018*). For the simultaneous delivery of the complete CRISPR machinery, we constructed a baculovirus-based silencing (BVSi) vector containing a CMV promoter-driven dCas9-KRAB-MeCP2-T2A-GFP expression cassette and a U6 promoter-driven gRNA expression cassette (*Figure 4—figure supplement 1b*). Multiple guides targeting exons 3 and 5 of *BEST1* were screened by nuclease surveyor assay, and the most efficient ones along with a non-specific scramble guide were individually constructed into the BVSi backbone for virus production. The resultant *BEST1*-targeting (BVSi 3–8 and BVSi 5–4) and control (BVSi-Ctrl) viruses were used to infect WT hPSC-RPE cells. Immunoblotting showed a better BEST1 knockdown efficiency of the BVSi 3–8 virus compared to the BVSi 5–4 virus (*Figure 4—figure supplement 1c*). Consistently, $Ca^{2+}$-dependent $Cl^-$ current from BVSi 3–8 infected cells was more effectively diminished compared to that from BVSi 5–4 infected cells at 1.2 μM $[Ca^{2+}]_i$ (*Figure 5a*), where RPE cells display the peak $Cl^-$ current amplitude. These results indicate a high silencing efficacy of the BVSi 3–8 design, which was used for later steps of the silencing/augmentation strategy.

For augmentation of WT BEST1 in the presence of BVSi 3–8, we generated baculovirus bearing a wobble WT BEST1-mCherry resistant to the recognition by gRNA 3–8 (*Figure 4—figure supplement 1c*). When wobble WT BEST1-mCherry was co-expressed, the diminished $Ca^{2+}$-dependent $Cl^-$ current in BVSi 3–8 treated WT hPSC-RPE cells was fully rescued at 1.2 μM $[Ca^{2+}]_i$ (*Figure 5b*), validating our silencing/augmentation system in WT hPSC-RPE cells. To test this strategy for the rescue of gain-of-function mutations, we carried out the same set of experiments in $BEST1^{I205T/WT}$ and $BEST1^{Y236C/WT}$ hPSC-RPE cells. Remarkably, the endogenous BEST1 protein was diminished with BVSi 3–8 treatment (*Figure 4—figure supplement 1d*) in the mutant hPSC-RPE cells, concomitant with abolished $Ca^{2+}$-dependent $Cl^-$ currents in these cells at 1.2 μM $[Ca^{2+}]_i$ (*Figure 5c–d*), while co-expression of the wobble WT BEST1-mCherry restored $Cl^-$ currents to the WT levels at all tested $[Ca^{2+}]_i$s (*Figure 5c–f*, *Figure 4—figure supplement 1d*), providing a proof-of-concept for the cure of bestrophinopathies associated with *BEST1* gain-of-function mutations.

## Discussion

In this study, we compared the influence of 10 patient-derived *BEST1* mutations, including one autosomal recessive mutation, six autosomal dominant loss-of-function mutations, and three autosomal dominant gain-of-function mutations, on the channel activity of BEST1 in transiently transfected HEK293 cells. Although the recessive and gain-of-function mutations indeed exhibited their expected recessive and dominant behaviors, respectively, the autosomal dominant loss-of-function mutations only dominated over the WT BEST1 at a superior 4:1 ratio, but not at a canonical 1:1 ratio

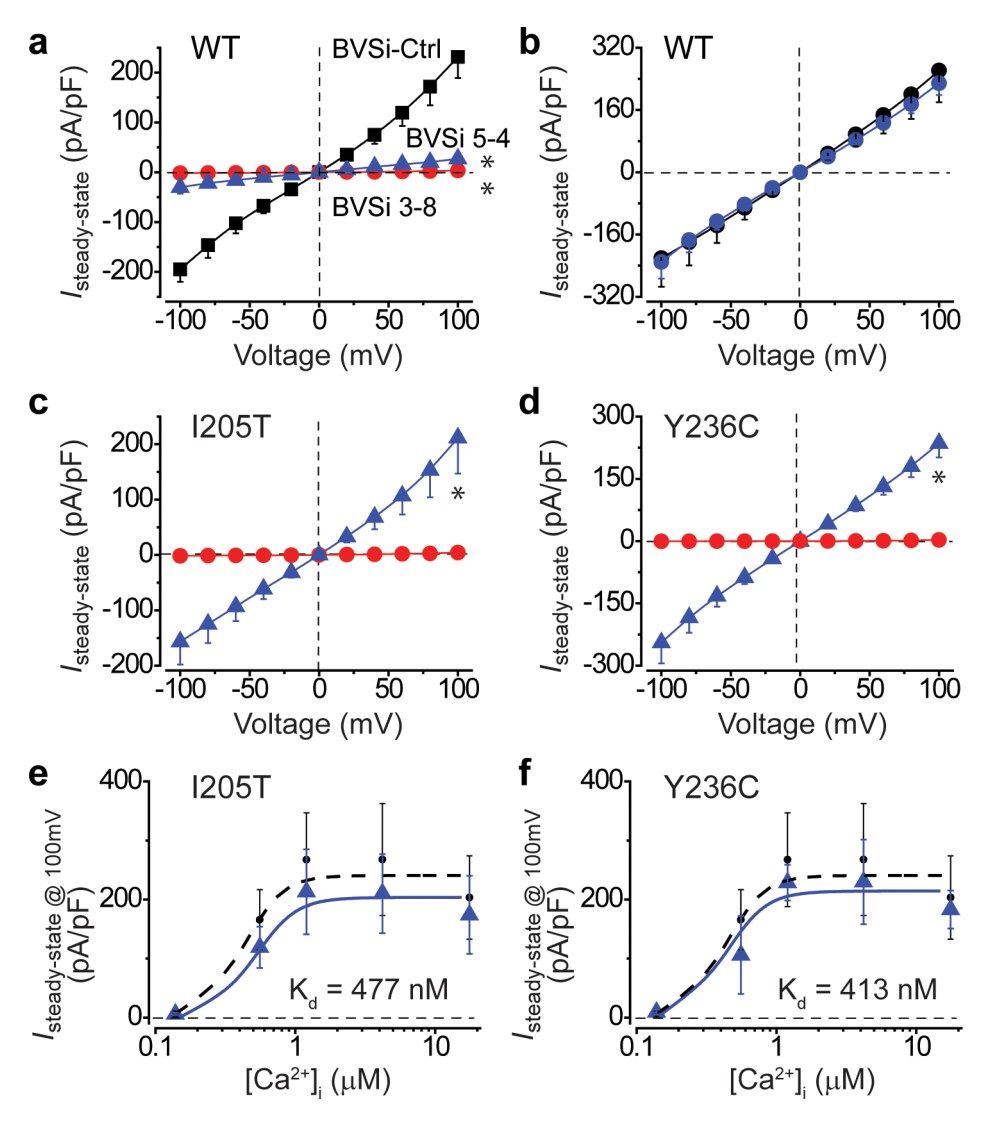

**Figure 5.** Knockdown and rescue of BEST1 gain-of-function mutations in hPSC-RPE cells. (**a**) Population steady-state current density-voltage relationships in WT hPSC-RPE cells treated with BVSi-Ctrl (black) compared to those in BVSi 3–8 (red) or BVSi 5–4 (blue) treated cells, at 1.2 μM $[Ca^{2+}]_i$, n = 5–17 for each point. *p<0.05 (8.3 × 10$^{-7}$ for BVSi 3–8 and 1.6 × 10$^{-6}$ for BVSi 5–4) compared to BVSi-Ctrl treated cells, using two-tailed unpaired Student's $t$ test. (**b**) Population steady-state current density-voltage relationships in WT hPSC-RPE cells treated with BVSi 3–8 plus wobble WT BEST1 (blue) compared to those in untreated cells (black), at 1.2 μM $[Ca^{2+}]_i$, n = 5–6 for each point. (**c–d**) Population steady-state current density-voltage relationships in $BEST1^{I205T/WT}$ (**c**) or $BEST1^{Y236C/WT}$ (**d**) hPSC-RPE cells treated with BVSi 3–8 alone (red), or BVSi 3–8 plus wobble WT BEST1 (blue), at 1.2 μM $[Ca^{2+}]_i$, n = 5–9 for each point. *p<0.05 (3.8 × 10$^{-3}$ for I205T and 2.7 × 10$^{-4}$ for Y236C) compared to cells treated with BVSi 3–8 alone, using two-tailed unpaired Student's $t$ test. (**e–f**) Steady-state current densities recorded at +100 mV plotted vs. $[Ca^{2+}]_i$ in $BEST1^{I205T/WT}$ (**e**) or $BEST1^{Y236C/WT}$ (**f**) hPSC-RPE cells treated with BVSi 3–8 plus wobble WT BEST1 (blue) compared to those in untreated WT hPSC-RPE (black), n = 5–6 for each point. The plots were fitted to the Hill equation. All error bars in this figure represent s.e.m.

(*Figure 1*). As the majority of the over 250 documented *BEST1* disease-causing mutations are autosomal dominant and display loss-of-function when tested *in vitro*, our results indicate an important role of allele-specific epigenetic control in the development of bestrophinopathies. In strong support of this finding, imbalanced transcription of the two endogenous *BEST1* alleles was detected in donor-derived iPSC-RPE and native RPE cells (*Table 1*), consistent with the previous observation

that *BEST1* is one of the inherited retinal disease genes with AEI in the human retinal transcriptome (*Llavona et al., 2017*).

AEI has been proven to be a common phenomenon in mammals (*Yan et al., 2002b*). An SNP array-based survey of 602 human genes discovered that more than half of the genes display AEI (*Lo et al., 2003*), while a separate study analyzing the mouse transcriptome revealed that ~20% of genes are prone to AEI in a tissue-specific manner (*Pinter et al., 2015*). Moreover, AEI of somatic mutations has been well documented in the context of cancer (*Bielski et al., 2018*; *Rhee et al., 2017*; *Yan et al., 2002a*; *Bielski and Taylor, 2021*), representing an important mechanism of tumorigenesis. However, the implication of AEI in monogenic diseases is poorly understood. To our knowledge, the linkage between an inherited missense mutation and AEI in pathogenesis has not been established yet. Our results suggest that bestrophinopathies caused by autosomal dominant mutations of *BEST1* may serve as a paradigm to address the influence of AEI in Mendelian disorders.

Conventionally, *BEST1* autosomal dominant mutations are identified when the mutation is present on just one of the two *BEST1* alleles in a bestrophinopathy patient. However, this classification only takes the genomic gene dosage into account but neglects the allelic transcription/expression level. The six autosomal dominant loss-of-function mutations tested in this study all behave recessively in HEK293 cells when co-expressed with the WT BEST1 at a 1:1 ratio, whereas the significantly decreased BEST1 channel activity in patient-derived iPSC-RPE cells is associated with a higher transcription level of the mutant allele compared to the WT counterpart, reflecting a dominant-negative effect rather than a canonical dominant effect. Therefore, we anticipate that a portion of the bestrophinopathy-causing mutations previously classified as autosomal dominant are *de facto* recessive and exhibit a dominant-negative phenotype when their expression outweighs that of the WT allele *in vivo*. This is in line with our previous finding that gene augmentation is sufficient to rescue *BEST1* loss-of-function mutations regardless of their inheritance patterns (*Ji et al., 2019b*), and provides an explanation for incomplete penetrance and variable clinical expressivity in patients bearing the same *BEST1* mutations (*Sodi et al., 2012*; *Cohn et al., 2011*; *Arora et al., 2016*).

BEST1's intrinsic functionality as a CaCC, physiological localization in RPE, and pathological relevance to retinal degenerative bestrophinopathies strongly suggest that BEST1 is the primary CaCC in RPE. Consistent with this idea, we previously reported an indispensable role of BEST1 in generating $Ca^{2+}$-dependent $Cl^-$ currents in donor-derived iPSC-RPE cells (*Li et al., 2017*). However, other candidates, including TMEM16A and TMEM16B, have also been proposed to be the physiological CaCC(s) in porcine or mouse RPE and the human RPE-derived ARPE-19 cells (*Schreiber and Kunzelmann, 2016*; *Keckeis et al., 2017*). Our results from isogenic knockout hPSC-RPE cells showed that $Ca^{2+}$-dependent $Cl^-$ currents were diminished in $BEST1^{-/-}$ cells, and remained intact in $TMEM16A^{-/-}$, $TMEM16B^{-/-}$, or $LRRC8A^{-/-}$ cells (*Figure 3*). Therefore, we conclude that BEST1 is the *bona fide* CaCC in human RPE.

We previously established a 'disease-in-a-dish' model, in which skin fibroblasts collected from the carriers of different *BEST1* mutations were reprogrammed into iPSC lines, and then differentiated into the corresponding iPSC-RPE cells for functional studies (*Figure 3a*, *Figure 3—figure supplement 3a–c*; *Li et al., 2017*; *Kittredge et al., 2018*). This iPSC-RPE- based model retains the patients' genetic background and thus has direct relevance to *BEST1*-associated retinal disorders, but is limited by the availability of patient samples. For instance, some *BEST1* mutations are rarer than others, and the carrier(s) may not be willing or logistically feasible to provide a sample. Here, we expanded the scope of our 'disease-in-a-dish' model based on an engineered hPSC line (H1-iCas9), which allows convenient introduction of desired *BEST1* mutations via the CRISPR/Cas9-mediated genome editing technique, generating isogenic hPSC lines that can be differentiated into isogenic hPSC-RPE cells (*Figures 3–4*). Importantly, almost identical $Ca^{2+}$-dependent $Cl^-$ currents were recorded from $BEST1^{WT/WT}$ hPSC-RPE compared to those from $BEST1^{WT/WT}$ iPSC-RPE (*Figure 3a*), validating hPSC-RPE as a versatile tool to model *BEST1* mutations.

As the BEST1 channel is a pentameric assembly, the number of mutant protomers required for displaying a phenotype could theoretically be 1, 2, 3, 4, or 5. Interestingly, five subtypes of bestrophinopathies have been documented, implying a potential correlation between the 'power' of the mutations and the resultant diseases. Supporting this hypothesis, ARB is specifically caused by *BEST1* autosomal recessive mutations, which represent the 'weakest' class that requires five mutant protomers in a channel pentamer to be phenotypic (*Figure 1h*, *Figure 1—figure supplement 1*). On the other hand, gain-of-function mutations such as D203A, I205T, and Y236C represent the

'strongest' class, which predominates over the WT BEST1 even at a 1:4 ratio (presumably one protomer per channel, *Figure 2a–c* and *Figure 2—figure supplement 1b*), although it remains unclear if they are specifically linked to a certain type of bestrophinopathy. Autosomal dominant loss-of-function mutations likely represent the 'intermediate' classes, which require 2–4 protomers in a BEST1 channel to display the mutant phenotype. For instance, the six loss-of-function mutations tested in this study (A10T, R218H, L234P, A243T, Q293K, and D302A) may represent the 4-mutant-protomer class as they are only dominant-negative at a 4:1 ratio with the WT in HEK293 cells, while Y85H, R92C, R218S, and G299E may represent the 2/3-mutant-protomer class(es), as they were previously shown to be dominant over the WT at a 1:1 ratio in HEK293 cells (*Sun et al., 2002*). However, the endogenous BEST1 mutant to WT molecule ratio in the RPE of bestrophinopathy patients with autosomal dominant mutations is still unknown, due to the lack of a quantitative approach to distinguish BEST1 missense variants from the WT counterpart at the protein level.

All three *BEST1* gain-of-function mutations in this study exhibit a strong dominant effect, suppressing the WT even at a 1:4 ratio (*Figure 2a–c*, *Figure 2—figure supplement 1b*). This suggests that for effective gene augmentation therapy, the total level of WT BEST1 protein, supplied both endogenously and exogenously, must be at least four folds higher than that of the endogenous mutant BEST1. However, we showed that even with a CMV promoter, which produces an apparently higher level of exogenous BEST1 protein compared to that of endogenous BEST1, the gain-of-function phenotype in $BEST1^{I205T/WT}$ and $BEST1^{Y236C/WT}$ hPSC-RPE cells cannot be rescued (*Figure 4b and e*, *Figure 4—figure supplement 1a*). Therefore, it seems impractical to rescue *BEST1* gain-of-function mutations by gene augmentation alone, especially considering that clinical applications may require the use of the native *BEST1* promotor, which is presumably not as strong as the CMV promotor. Structurally, the three gain-of-function mutations (D203A, I205T, and Y236C) are located at or in a close proximity to the neck (I76, F80, and F84) or the aperture (I205) of the channel (*Figure 1—figure supplement 2*), and are involved in the opening of at least one of these two $Ca^{2+}$-dependent gates (*Ji et al., 2019a*; *Zhang et al., 2018*). For instance, the I205T mutation, replacing a bulky isoleucine with a smaller side-chained threonine at the aperture (*Figure 1—figure supplement 2*), causes a $Ca^{2+}$-independent "leak" due to enlargement of the channel constriction (*Figure 2b*, *Figure 4a-c*; *Ji et al., 2019a*). By contrast, loss-of-function mutations are located in various regions of the channel (*Ji et al., 2019b*).

There are two common strategies to overcome the strong dominant effect of gain-of-function mutations: (1) specific suppression of the endogenous mutant allele and (2) non-selective suppression of both endogenous alleles in combination with WT gene augmentation. We applied the latter approach in this study using a CRISPR/Cas9-based gene silencing vector (BVSi) to suppress the endogenous *BEST1* expression (*Tsai, 2018*). As the *BEST1* genomic locus recognized by our BVSi does not have any reported disease-causing mutations or polymorphisms, this BVSi design is universally suited for *BEST1* silencing in bestrophinopathy patients no matter where their mutations are located within the gene. Notably, although gene augmentation alone is readily sufficient to rescue loss-of-function mutations (*Ji et al., 2019b*), simultaneously suppressing the endogenous BEST1 does not interfere with the functional restoration. Therefore, our silencing plus augmentation combination strategy can potentially be utilized for the treatment of all bestrophinopathies.

## Materials and methods

**Key resources table**

| Reagent type (species) or resource | Designation | Source or reference | Identifiers | Additional information |
|---|---|---|---|---|
| Strain, strain background (*Escherichia coli*) | HST08 (Stellar cells) | TaKaRa | 636766 | Chemical competent cells |
| Cell line (*Spodoptera frugiperda*) | Sf9 | Thermo Fisher Scientific | RRID:CVCL_0549 | Insect cell line for baculovirus production |

*Continued on next page*

*Continued*

| Reagent type (species) or resource | Designation | Source or reference | Identifiers | Additional information |
|---|---|---|---|---|
| Cell line (*Homo sapiens*) | HEK293 | ATCC | RRID:CVCL_0045 | Embryonic kidney cells |
| Cell line (*Homo sapiens*) | H1-iCas9 | Sloan Kettering Institute, **González et al., 2014** | | Embryonic stem cell line with an inducible CRISPR cassette |
| Cell line (*Homo sapiens*) | H1-iCas9 BEST1$^{-/-}$ | This paper | | BEST1$^{-/-}$ knockout generated from the H1-iCas9 line |
| Cell line (*Homo sapiens*) | H1-iCas9 TMEM16A$^{-/-}$ | This paper | | TMEM16A$^{-/-}$ knockout generated from the H1-iCas9 line |
| Cell line (*Homo sapiens*) | H1-iCas9 TMEM16B$^{-/-}$ | This paper | | TMEM16B$^{-/-}$ knockout generated from the H1-iCas9 line |
| Cell line (*Homo sapiens*) | H1-iCas9 LRRC8A$^{-/-}$ | This paper | | LRRC8A$^{-/-}$ knockout generated from the H1-iCas9 line |
| Cell line (*Homo sapiens*) | H1-iCas9 BEST1$^{I205T/WT}$ | This paper | | BEST1$^{I205T/WT}$ knock-in generated from the H1-iCas9 line |
| Cell line (*Homo sapiens*) | H1-iCas9 BEST1$^{Y236C/WT}$ | This paper | | BEST1$^{Y236C/WT}$ knock-in generated from the H1-iCas9 line |
| Biological sample (*Homo sapiens*) | RPE cells | **Li et al., 2017** | | Human RPE cells from a post-mortem donor |
| Biological sample (*Homo sapiens*) | iPSC-RPE cells | **Ji et al., 2019a** | | iPSC-RPE cells derived from patient skin cells |
| Antibody | Anti- RPE65 (Mouse monoclonal) | Novus Biologicals | Cat#: NB100-355, RRID:AB_10002148 | WB (1:1,000) |
| Antibody | Anti-CRALBP (mouse monoclonal) | Abcam | Cat#: ab15051, RRID:AB_2269474 | WB (1:500) |
| Antibody | Anti- BEST1 (mouse monoclonal) | Novus Biologicals | Cat#: NB300-164, RRID:AB_10003019 | WB (1:500) |
| Antibody | Anti-β-actin (rabbit polyclonal) | Abcam | Cat#: ab8227, RRID:AB_2305186 | WB (1:2,000) |
| Antibody | Anti- 6xHis (rabbit polyclonal) | Thermo Fisher Scientific | Cat#: PA1-983B, RRID:AB_1069891 | WB (1:1,000) |
| Antibody | Anti-Myc (rabbit polyclonal) | Thermo Fisher Scientific | Cat#: PA1-981, RRID:AB_325961 | WB (1:1,000) |
| Antibody | IRDye 680RD anti-mouse IgG (goat polyclonal) | LI-COR Biosciences | Cat#: 925–68070, RRID:AB_2651128 | WB (1:10,000) |
| Antibody | IRDye 800CW anti-rabbit IgG (donkey polyclonal) | LI-COR Biosciences | Cat#: 925–32213, RRID:AB_2715510 | WB (1:10,000) |
| Recombinant DNA reagent | pEG BacMam | **Goehring et al., 2014** | | Baculoviral vector for gene expression |
| Recombinant DNA reagent | pBacMam-BEST1-GFP (plasmid) | **Li et al., 2017** | | To express exogenous BEST1 in HEK293 cells |
| Recombinant DNA reagent | pBacMam-BEST1-mCherry (plasmid) | This paper | | Made from pEG BacMam by inserting BEST1-mCherry |
| Recombinant DNA reagent | dCas9-KRAB-MeCP2 (plasmid) | Addgene | RRID:Addgene_110821 | Improved dCas9 repressor-dCas9-KRAB-MeCP2 |
| Recombinant DNA reagent | pSpCas9(BB)—2A-GFP (PX458) (plasmid) | Addgene | RRID:Addgene_48138 | Cas9 from *Streptococcus pyogenes* with 2A-EGFP, and cloning backbone for sgRNA |

*Continued on next page*

*Continued*

| Reagent type (species) or resource | Designation | Source or reference | Identifiers | Additional information |
|---|---|---|---|---|
| Recombinant DNA reagent | BVSi 5–4-GFP (plasmid) | This paper | | Made from pEG BacMam, dCas9-KRAB-MeCP2 and pSpCas9(BB)—2A-GFP, for *BEST1* silencing |
| Recombinant DNA reagent | BVSi 3–8-GFP (plasmid) | This paper | | Made from pEG BacMam, dCas9-KRAB-MeCP2 and pSpCas9(BB)—2A-GFP, for *BEST1* silencing |
| Recombinant DNA reagent | BVSi ctrl-GFP (plasmid) | This paper | | Made from pEG BacMam, dCas9-KRAB-MeCP2 and pSpCas9(BB)—2A-GFP, serving as a control for *BEST1* silencing |
| Sequence-based reagent | hBest1-I205T-ssDNA | This paper | Knock-in ssDNA template | GCCCTGGGTGTGGTTTGC CAACCTGTCAATGAAGGC GTGGCTTGGAGGTCGAAT TCGGGACCCTACCCTGCTC CAGAGCCTGCTGAACGTG AGCCCACTGTACAGACAG GGCTGCCGCAG |
| Sequence-based reagent | hBest1-Y236C-ssDNA | This paper | Knock-in ssDNA template | TCAGTGTGGACACCTGTA TGCCTACGACTGGATTA GTATCCCACTGGTGTGTA CACAGGTGAGGACTAGTC TGGTGAGGCTGCCCTTTT GGGAAACTGAGGCTAGAA GGACCAAGGAAGC |
| Commercial assay or kit | CytoTune-iPS 2.0 Sendai reprogramming kit | Thermo Fisher Scientific | Cat#: A16517 | To generate iPSC |
| Commercial assay or kit | In-Fusion HD Cloning | Clontech | Clontech:639647 | For molecular cloning |
| Commercial assay or kit | PolyJet *In Vitro* DNA Transfection Reagent | SignaGen Laboratories | SL100688 | For cell transfection |
| Software, algorithm | Patchmaster | HEKA | RRID:SCR_000034 | Patch clamp data collection and analysis |
| Software, algorithm | PyMOL | PyMOL | RRID:SCR_000305 | Structural analysis |

## Generation of human iPSC

The CytoTune-iPS 2.0 Sendai Reprogramming Kit (Thermo Fisher Scientific, A16517) was used to reprogram donor-provided skin fibroblasts into pluripotent stem cells (iPSC). Immunocytofluorescence assays were carried out following the previously published protocol to score iPSC pluripotency (*Li et al., 2016*). The iPSC cells from all the subjects enrolled in this study were characterized by detecting four standard pluripotency markers (SSEA4, Tra-1–60, SOX2, and Nanog). Nuclei were detected by Hoechst staining. All iPSC lines were passaged every 3–6 days while maintained in mTeSR-1 medium (STEMCELL Technologies, 85850). The morphology and nuclear/cytoplasmic ratio were closely monitored to ensure the stability of the iPSC lines. All the iPSC lines were sent for karyotyping by G-banding to verify genome integrity at Cell Line Genetics (Madison, WI).

## Differentiation of iPSC and hPSC lines into RPE cells

iPSC and hPSC lines were cultured to confluence in six-well culture dishes pretreated with 1:50 diluted matrigel (CORNING, 356230). For the first 14 days, the differentiation medium consisted of Knock-Out (KO) DMEM (Thermo Fisher Scientific, 10829018), 15% KO serum replacement (Thermo Fisher Scientific, 10829028), 2 mM glutamine (Thermo Fisher Scientific, 35050061), 50 U/ml penicillin-streptomycin (Thermo Fisher Scientific, 10378016), 1% nonessential amino acids (Thermo Fisher Scientific, 11140050), and 10 mM nicotinamide (Sigma-Aldrich, N0636). During days 15–28 of

differentiation, the differentiation medium was supplemented with 100 ng/ml human Activin-A (PeproTech, 120–14). From day 29 on, the differentiation medium without Activin-A supplementation was used again until differentiation was completed. After roughly 8–10 weeks, dispersed pigmented flattened clusters were formatted and manually picked to matrigel-coated dishes. These cells were kept in RPE culture medium as previously described (Maminishkis et al., 2006). It takes another 6–8 weeks in culture for them to form a functional monolayer, which would be ready for function assays. In addition to well-established classical mature RPE markers (Bestrophin1, CRALBP, and RPE65), two more markers (PAX6 and MITF) were also used to validate the RPE fate of the cells. All iPSC-RPE cells in this study were at passage 1. DNA sequencing was used to verify genomic mutations in the mutant iPSC-RPE cells.

## Cell lines

HEK293 cells were purchased from ATCC. As HEK293 is on the International Cell Line Authentication Committee's list of commonly misidentified cell lines, the cells used in this study were authenticated by short tandem repeat DNA profiling and tested negative for mycoplasma contamination. The culture medium was DMEM (4.5 g/l glucose, Corning 10013CV) supplemented with 100 μg/ml penicillin-streptomycin and 10% fetal bovine serum.

H1-iCas9 cells were purchased from the Stem Cell Research Facility of Memorial Sloan Kettering Cancer Center. The culture medium was mTeSR1 with supplement (STEMCELL Technologies, 85850).

## Electrophysiology

An EPC10 patch clamp amplifier (HEKA Electronics) controlled by Patchmaster (HEKA) was utilized to conduct whole-cell recordings 24–72 hr after splitting of RPE cells or transfection of HEK293 cells. Micropipettes were pulled and fashioned from 1.5 mm thin-walled glass with filament (WPI Instruments) and filled with internal solution containing (in mM): 130 CsCl, 10 EGTA, 1 MgCl$_2$, 2 MgATP (added fresh), 10 HEPES (pH 7.4, adjusted by CsOH), and CaCl$_2$ to obtain the desired free Ca$^{2+}$ concentration (maxchelator.stanford.edu/CaMgATPEGTA-TS.htm). Series resistance was usually 1.5–2.5 MΩ. No electronic series resistance compensation was used. External solution contained (in mM): 140 NaCl, 15 glucose, 5 KCl, 2 CaCl$_2$, 1 MgCl$_2$, and 10 HEPES (pH 7.4, adjusted by NaOH). Solution osmolarity was between 310 and 315. A family of step potentials (−100 to +100 mV from a holding potential of 0 mV) were used to generate I-V curves. Currents were sampled at 25 kHz and filtered at 5 or 10 kHz. Traces were acquired at a repetition interval of 4 s (Yang et al., 2014b). All experiments in this study were carried out at ambient temperature (23 ± 2°C).

## Immunoblotting

Cell pellets were extracted by the M-PER mammalian protein extraction reagent (Thermo Fisher Scientific, 78501) supplemented with proteinase inhibitors (Roche, 04693159001), and the protein concentration was quantified by a Bio-Rad protein reader. After denaturing at 95°C for 5 min, the samples (20 μg) were run on 4–15% gradient SDS-PAGE gel at room temperature, and wet transferred onto nitrocellulose membrane at 4°C. The membranes were incubated in blocking buffer containing 5% (w/v) non-fat milk for 1 hr at room temperature and subsequently incubated overnight at 4°C in blocking buffer supplemented with primary antibody. Primary antibodies against the following proteins were used: CRALBP (1:500 Abcam, ab15051), RPE65 (1:1,000 Novus Biologicals, NB100-355), β-Actin (1:2,000 Abcam, ab8227), BEST1 (1:500 Novus Biologicals, NB300-164), His (1:1,000 Fisher Scientific, PA1983B), and Myc (1:1,000 Fisher Scientific, PA1981). Fluorophore-conjugated mouse and rabbit secondary antibodies (LI-COR Biosciences, 925–68070 and 925–32213, respectively) were used at a concentration of 1:10,000 and an incubation time of 1 hr at room temperature, followed by infrared imaging.

## Immunoprecipitation

HEK293 cells cultured on 6 cm dishes were co-transfected with pBacMam-BEST1(WT)-CFP-Myc and pBacMam-BEST1(mutant or WT)-YFP-His at 1:1 ratio using PolyJet In Vitro DNA Transfection Reagent (SignaGen Laboratories, SL100688) following the manufacturer's standard manual. Forty-eight hours post transfection, cells were harvested by centrifugation at 1000 × g for 5 min at room

temperature. Cell pellets were lysed in pre-cooled lysis buffer (150 mM NaCl, 50 mM Tris, 0.5% IGE-PAL CA-630, pH 7.4) supplemented with protease inhibitor cocktails (Roche, 04693159001) for 30 min on ice, and then centrifuged at 13,000 rpm for 12 min at 4°C. The supernatant (300 µg) was collected and mixed with 2 µg Myc monoclonal antibody (Thermo Fisher Scientific, MA1-980). After rotating overnight at 4°C, the mixture was incubated with Dynabeads M-280 sheep anti-mouse IgG (Thermo Fisher Scientific, 11202D) for 5 hr at 4°C. After thorough washing of the beads, bound fractions were eluted in 1× SDS sample buffer (Biorad, 1610747) by heating for 10 min at 75°C. Proteins were then resolved by SDS-PAGE and analyzed by immunoblotting.

### Baculovirus production and transduction

BacMam baculovirus bearing BVSi 5–4-GFP, BVSi 3–8-GFP, BVSi-Ctrl-GFP, or wobble BEST1-mcherry were generated in-house as previously described (*Goehring et al., 2014*). For transduction, the viruses were added to the culture medium of freshly split hPSC-RPE cells.

### Molecular cloning

Point mutations in BEST1 were made by site-directed mutagenesis PCR with the In-fusion Cloning Kit (Clontech). All constructs were fully sequenced.

### Measuring allelic transcription level

Total RNA was extracted from cell pellets with the PureLink RNA Mini Kit (ThermoFisher, 12183020) and subjected to cDNA synthesis using the RevertAid First Strand cDNA synthesis kit (Thermo Fisher K1621). The resultant cDNA was used as the template for PCR amplification of the target *BEST1* regions that contain mutations/polymorphisms, and the PCR products were sub-cloned using the TOPO Cloning Kit (Thermo Fisher, 451245) for sequencing.

### Knockout/knock-in in H1-iCas9 cells

Doxycycline (2 µg/ml) was supplemented to the culture medium to induce Cas9 expression and maintained in the medium for 3 days. Twenty-four hours post doxycycline addition, the cells were transfected with gRNA (+ssDNA for knock-in) as previously described (*Zhu et al., 2014*). After recovery to ~50% confluency, the cells were lifted by TrypLE (Thermo Fisher, 12604013) treatment, and seeded to $2 \times 10\ cm^2$ fresh plates at 1000 and 2000 cells/plate, respectively. Ten to 12 days later, single colonies became visible and were picked into individual wells on a 96-well plate. After amplification, each single colony was subjected for genotyping by Sanger sequencing.

For the knockout of BEST1, TMEM16A, TMEM16B, and LRRC8A, gRNAs were designed to target the N-terminal portion of the coding genomic sequences, such that all or most of the transmembrane domain is eliminated in the residual translated product (if it exists), rendering it functionally null.

### gRNA design for CRISPR/Cas9-mediated gene editing/silencing

The gRNAs were designed using online software (http://www.IDTdna.com) and are summarized in *Figure 3—source data 1*.

### Transfection

Twenty to 24 hr before transfection, HEK293 cells were lifted by incubation with 0.25% trypsin at room temperature for 5 min and split into new 3.5 cm culture dishes at ~50% confluency. Plasmids (1 µg) bearing the WT BEST1 or desired mutant were transfected using PolyJet transfection reagent (SignaGen SL100688). The transfection mix was removed after 4–8 hr, and cells were rinsed with PBS once and cultured in supplemented DMEM. Twenty-four hours post transfection, cells were lifted again by trypsin treatment and split onto fibronectin-coated glass coverslips for patch clamp (*Yang et al., 2013*).

### Electrophysiological data and statistical analyses

Patch clamp data were analyzed off-line with Patchmaster (HEKA), Microsoft Excel, and Origin. Statistical analyses were conducted using built-in functions in Origin. For comparisons between two

groups, statistically significant differences between means (p<0.05) were determined using Student's *t* test. Data are presented as means ± s.e.m (*Yang et al., 2007*).

## Homology modeling of human BEST1

A homology model for BEST1 was generated using the Swiss-Model server from the chicken BEST1 crystal structure (*Kane Dickson et al., 2014*). The structural figure was made in PyMOL.

## Human samples

Skin biopsy samples were obtained from a healthy control donor and patients, and processed and cultured as previously described (*Li et al., 2016*). For these procedures, all of which were approved by Columbia University Institutional Review Board (IRB) protocol AAAF1849, the donors provided written informed consent. All methods were performed in accordance with the relevant regulations and guidelines. Donor native RPE was isolated from human autopsy eye shell purchased from the Eye-Bank for Sight Restoration (New York, NY, 10005).

## Acknowledgements

We thank the Unrestricted Grant to the Department of Ophthalmology, Columbia University, from Research to Prevent Blindness (RPB). YS was supported by National Key R and D Program of China (2017YFE0103400). The Jonas Children's Vision Care was supported by NIH grants (EY028758, CA013696, EY030580, OD020351, EY027285, EY019007, EY018213, EY024698, EY026682, AG050437), the Schneeweiss Stem Cell Fund, New York State (SDHDOH01-C32590GG-3450000), the Foundation Fighting Blindness New York Regional Research Center Grant (PPA-1218–0751-COLU), Nancy and Kobi Karp, the Crowley Family Funds, the Rosenbaum Family Foundation, Alcon Research Institute, the Gebroe Family Foundation, and the RPB Physician-Scientist Award. TY was supported by NIH grants (GM127652, EY028758), the Irma T Hirschl/Monique Weill-Caulier Research Award, and Columbia University Faculty Recruitment Award.

## Additional information

### Competing interests

Yu Zhang, Tingting Yang: Provisional Patent Application No. 63/174,090. Stephen H Tsang: Provisional Patent Application No. 63/174,090. Stephen H Tsang has received financial benefits from Spark Therapeutics and research support from Abeona Therapeutics, Inc and Emendo. The other authors declare that no competing interests exist.

### Funding

| Funder | Grant reference number | Author |
| --- | --- | --- |
| National Key R&D Program of China | 2017YFE0103400 | Yin Shen |
| National Institutes of Health | EY028758 | Stephen H Tsang |
| National Institutes of Health | CA013696 | Stephen H Tsang |
| National Institutes of Health | EY030580 | Stephen H Tsang |
| National Institutes of Health | OD020351 | Stephen H Tsang |
| National Institutes of Health | EY027285 | Stephen H Tsang |
| National Institutes of Health | EY019007 | Stephen H Tsang |
| National Institutes of Health | EY018213 | Stephen H Tsang |
| National Institutes of Health | EY024698 | Stephen H Tsang |
| National Institutes of Health | EY026682 | Stephen H Tsang |
| National Institutes of Health | AG050437 | Stephen H Tsang |
| Foundation Fighting Blindness | PPA-1218-0751-COLU | Stephen H Tsang |

| Schneeweiss Stem Cell Fund | SDHDOH01-C32590GG-3450000 | Stephen H Tsang |
| --- | --- | --- |
| Nancy & Kobi Karp | | Stephen H Tsang |
| Crowley Family Funds | | Stephen H Tsang |
| Rosenbaum Family Foundation | | Stephen H Tsang |
| Alcon Research Institute | | Stephen H Tsang |
| Gebroe Family Foundation | | Stephen H Tsang |
| Research to Prevent Blindness | Physician-Scientist Award | Stephen H Tsang |
| National Institutes of Health | GM127652 | Tingting Yang |
| National Institutes of Health | EY028758 | Tingting Yang |
| Irma T. Hirschl/Monique Weill-Caulier Trust | Research Award | Tingting Yang |
| Columbia University | Faculty Recruitment Award | Tingting Yang |

The funders had no role in study design, data collection and interpretation, or the decision to submit the work for publication.

### Author contributions

Qingqing Zhao, Data curation, Validation, Investigation, Methodology, Formal analysis; Yang Kong, Data curation, Validation, Investigation, Methodology, Writing - review and editing; Alec Kittredge, Data curation, Investigation, Writing - review and editing; Yao Li, Data curation; Yin Shen, Supervision, Writing - review and editing; Yu Zhang, Conceptualization, Data curation, Formal analysis, Supervision, Validation, Investigation, Methodology, Writing - original draft, Project administration, Writing - review and editing; Stephen H Tsang, Resources, Supervision, Funding acquisition, Validation, Writing - review and editing; Tingting Yang, Conceptualization, Resources, Formal analysis, Supervision, Funding acquisition, Validation, Investigation, Methodology, Writing - original draft, Project administration, Writing - review and editing

### Author ORCIDs

Alec Kittredge https://orcid.org/0000-0002-8140-9267
Yin Shen https://orcid.org/0000-0002-4201-3948
Tingting Yang https://orcid.org/0000-0002-5220-588X

### Decision letter and Author response

Decision letter https://doi.org/10.7554/eLife.67622.sa1
Author response https://doi.org/10.7554/eLife.67622.sa2

# Additional files

## Supplementary files
• Transparent reporting form

## Data availability
All data generated or analysed during this study are included in the manuscript and supporting files. Source data files have been provided for Figure 2 and Figure 3.

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
