## [Decision Letter]

**Acceptance summary:**

This study addresses the genetic basis for different "bestrophinopathies" which are a family of retinal diseases caused by gene mutations in BEST1, an ion channel. The experiments are methodically planned and executed, the data are well-described and interpreted, and the results will be very interesting to a broad audience, including channel biophysicists and clinicians alike as they consider the physiological function of BEST1 and the development of therapies to treat Best's dystrophy.

**Decision letter after peer review:**

Thank you for submitting your article "Distinct expression requirements and rescue strategies for BEST1 loss- and gain-of-function mutations" for consideration by *eLife*. Your article has been reviewed by 3 peer reviewers, and the evaluation has been overseen by a Reviewing Editor and Richard Aldrich as the Senior Editor. The following individual involved in review of your submission has agreed to reveal their identity: Wallace B Thoreson (Reviewer #1).

Essential revisions:

1. The text states that Ca^2+^-independent currents in both HEK cells and iPSC-derived RPE cells expressing the I205T mutant gene were significantly larger than WT currents. However, the differences shown in the paper are extremely small and not very convincing. Is this truly a gain-of-function Ca^2+^-dependent Cl^-^ current? Especially since there was no change in Ca^2+^-dependent currents? In the rescue experiments of Figure 5, the generality of their conclusion that one must first silence the mutant gene would be more strongly supported if they tested the D203A mutant that shows a more appreciable increase in function. Do the authors have data with this mutant? What is the rationale for studying the I205T mutant rather than the D203A mutant? We hope you may have data in hand from the D203A mutant to further substantiate their conclusions, but if not we think you can answer this question without further experiments.

2. Please state explicitly in the text that the mutants studied in Table 1 are the loss-of-function mutants shown in Figure 2. We do not see it stated clearly in the text. In addition, it would be interesting to see the transcription levels for gain-of-function mutants if they are available.

3. Please provide some discussion on mutant transcription regulation of WT alleles in other diseases. Is this common? Uncommon?

4. There were many mutations tested in this manuscript, some tested dominant negative, some dominant, some recessive. Since there are structures of BEST1 and -2 and are similar in structure to each other, it is curious if the site of mutants reside closely in a region or close in space to each other on the 3D structure, especially in the pentameric form. If so, it would be helpful and intriguing to show that in a final figure; if they do not align close in space to each other, then stating that within the text would be beneficial.

5. The difference between iPSCs (used in figure 3) and hPSCs (used in figure 4) is not clear.

6. The authors might consider including a key in the figure, especially where four traces are included in a single I-V plot. Although all of the information is included in the figure legend, the reader might be able to more quickly understand figure without needing to go back and forth with the figure legend when examining the data.

7. The authors might also consider using labels within the figures to remind readers when the data are 1:1, 4:1 or 1:4 in figures 1-3 and where applicable. Although the text explains when the different ratios are used and the experiments are well motivated, it took us a second read to clarify these differences, which are important in the light of loss-of-function versus gain-of-function mutations.

*Reviewer #1:*

This study addresses the genetic basis for different "bestrophinopathies" which are a family of retinal diseases caused by BEST1 gene mutations. The authors tested gene dosage effects by expressing different levels of various loss-of-function and gain-of-function BEST1 mutants in HEK293 cells. Mutant Ca^2+^-activated Cl^-^ currents matched WT currents when dominant loss-of-function mutants were expressed at 1:1 ratio with WT BEST1 but were reduced when mutants were expressed at 4:1 ratio. Expression of loss-of-function mutant alleles in RPE cells differentiated from human iPSC cells with BEST1 mutations and in native RPE cells from a Best's patient showed 3-4 fold greater transcription of the mutant allele. These data suggest that these nominally autosomal dominant mutants actually behave like dominant-negative mutants. By comparison, an autosomal recessive mutation behaved as expected, exhibiting WT currents at both 1:1 and 4:1 ratios but diminished currents when the mutant was expressed in the absence of WT cDNA. Two gain of function mutants showed enhanced Ca^2+^-dependent Cl^-^ currents when expressed at a ratio of only 1:4 with WT cDNA. In RPE cells derived from human iPSCs, endogenous Ca^2+^-activated Cl^-^ currents were eliminated after knockout of BEST1 gene but not other potential candidates (e.g., TMEM16), providing further evidence in support of the hypothesis that BEST1 forms Ca^2+^-activated Cl^-^ channels in these cells. They then examined two gain-of-function mutants in derived RPE cells to test whether over-expressing WT genes could rescue these mutant cells. Unlike loss-of-function mutants, they found that rescue only occurred if they first silenced the mutant gene. This suggests a potential strategy for treating gain-of-function mutations.

The study design is clear and clean. The data quality is quite high, incorporating both molecular and electrophysiological studies of expression levels in two types of cells: HEK293 cells and iPSC-derived RPE cells. The results are generally clear and convincing. The potential impact for future treatment of Best's dystrophy is high.

*Reviewer #2:*

This manuscript is an elegant study that reports that BEST1 patient-derived loss-of-function and gain-of-function mutations require different mutant/WT molecule ratios in order to produce a phenotype. Through quantitative examination in first transiently transfected HEK cells then in patient-derived RPE cells and in an hPSC-RPE model system, the authors showed that six mutations in the BEST1 gene that were found associated with an autosomal dominant form of bestrophiopathy were affecting the function of the Ca++-dependent Cl^-^ channel (CaCC) in a dominant manner. In fact, these six mutations required a 4:1 ratio with WT BEST1 to exhibit the mutant phenotype, indicative of a recessive trait. This is an intriguing result, and correlates well with data they obtained from RT-PCR of mRNA taken from patient-derived iPSC-RPE clones. These data indicated the transcription level of the mutant allele is 3-4-fold higher than that of the WT allele in these patients' cells. In contrast, the authors showed that three autosomal dominant gain-of-function mutations all displayed a dominant behavior, even at a 1:4 ratio with the WT BEST1. Additionally, the authors confirmed the role of BEST1 as the genuine CaCC in RPE.

The manuscript is very well-written. The experiments are methodically planned and executed, the data are well-described and interpreted, and are very interesting to a broad audience.

*Reviewer #3:*

Background: Bestrophin 1 (BEST1) is a Ca^2+^-activated Cl^-^ implicated in a number of retinal degeneration disorders known as bestrophinopathies. There are 250 distinct BEST1 mutations linked to autosomal dominant disorders whereas autosomal recessive mutations are linked to autosomal recessive bestrophinopathy (ARB). Despite these findings, the pathological mechanisms behind these autosomal dominance or autosomal recessive mutations remains unclear. Here, the authors sought to determine whether autosomal dominant loss-of-function and gain-of-function BEST1 mutations influence channel activity differently in heterozygous carriers.

Synopsis: Using recordings of Best1 channels with disease causing mutations, the authors report that BEST1 loss-of-function and gain-of-function mutations require different mutant:WT ratios to manifest phenotypes. They also report that BEST1 gain-of-function mutations do not restore BEST1-dependent Cl^-^ currents whereas loss-of-function mutations get rescued. Finally, they report that the combination of gene augmentation and CRISPR/Cas9-mediated knockdown of BEST1 could be a potential treatment.

The experiments presented here are exciting and would be interesting to channel biophysicists and clinicians alike as they address an important question in relation to bestrophinopathies as well as the effects of mutant to wild-type ratio in disease manifestation.

---

## [Author Response]

Essential revisions:1. The text states that Ca^2+^-independent currents in both HEK cells and iPSC-derived RPE cells expressing the I205T mutant gene were significantly larger than WT currents. However, the differences shown in the paper are extremely small and not very convincing. Is this truly a gain-of-function Ca^2+^-dependent Cl^-^ current? Especially since there was no change in Ca^2+^-dependent currents? In the rescue experiments of Figure 5, the generality of their conclusion that one must first silence the mutant gene would be more strongly supported if they tested the D203A mutant that shows a more appreciable increase in function. Do the authors have data with this mutant? What is the rationale for studying the I205T mutant rather than the D203A mutant? We hope you may have data in hand from the D203A mutant to further substantiate their conclusions, but if not we think you can answer this question without further experiments.

The gain-of-function phenotype of I205T is exhibited under no or low Ca^2+^ conditions, where significantly larger currents are consistently recorded from both endogenous BEST1 in hPSC-RPE (*BEST1^I205T/WT^* vs. *BEST1^WT/WT^*: 73 ± 22 vs. 4 ± 1 pA/pF in the absence of Ca^2+^, 71 ± 26 vs. 6 ± 2 pA/pF in the presence of 139 nM Ca^2+^, Figure 4c) and exogenous BEST1 transiently expressed in HEK293 (I205T vs. WT: 13 ± 3 vs. 3 ± 0.4 pA/pF in the absence of Ca^2+^, Figure 2b and Figure 2‒figure supplement 1), representing a Ca^2+^-independent “leak” due to the enlarged aperture (I205, which is a Ca^2+^-dependent gate of the channel), caused by the I205T mutation. We have added a figure showing the locations of these mutations (Figure 1‒figure supplement 2), and a description paragraph in **Discussion** (Page 15).

We attempted to generate *BEST1^D203A/WT^*, *BEST1^I205T/WT^* and *BEST1^Y236C/WT^* hPSC cells at the same time, but unfortunately failed to obtain *BEST1^D203A/WT^* hPSC due to technical issues. We have been actively trouble shooting, but so far only have data from *BEST1^I205T/WT^* and *BEST1^Y236C/WT^* hPSC-RPE cells.

2. Please state explicitly in the text that the mutants studied in Table 1 are the loss-of-function mutants shown in Figure 2. We do not see it stated clearly in the text. In addition, it would be interesting to see the transcription levels for gain-of-function mutants if they are available.

As the reviewer suggested, we have added in the Table 1 legend that they are donor derived iPSC-RPEs carrying the same *BEST1* mutations as those analyzed in transiently expressed HEK293 in Figure 1.

We currently do not have any *BEST1* gain-of-function patient samples, so cannot access the transcription levels of these mutants under a pathologically relevant setting such as patient derived iPSC-RPE cells. The hPSC-RPE cells carrying gain-of-functions (*BEST1^I205T/WT^* and *BEST1^Y236C/WT^*) in the manuscript are both engineered from the same parental iCas9-H1 hPSC line, so their *BEST1* allelic transcription levels would unlikely inform the epigenetic control of *BEST1* in real patients.

3. Please provide some discussion on mutant transcription regulation of WT alleles in other diseases. Is this common? Uncommon?

As the reviewer suggested, we have added a paragraph of discussion on this point (Page 12).

4. There were many mutations tested in this manuscript, some tested dominant negative, some dominant, some recessive. Since there are structures of BEST1 and -2 and are similar in structure to each other, it is curious if the site of mutants reside closely in a region or close in space to each other on the 3D structure, especially in the pentameric form. If so, it would be helpful and intriguing to show that in a final figure; if they do not align close in space to each other, then stating that within the text would be beneficial.

As the reviewer suggested, we have added a figure (Figure 1‒figure supplement 2) showing the locations of the examined mutations in a BEST1 channel homology model. The three gain-of-function mutations (D203A, I205T and Y236C) are at or in a close proximity to the neck or the aperture (composed of I76/F80/F84 and I205, respectively, both are Ca^2+^-dependent gates of the channel), and are involved in the opening of at least one of the gates. By contrast, loss-offunction mutations are located in various regions of the channel. We have added this information in **Discussion** (Page 15).

5. The difference between iPSCs (used in figure 3) and hPSCs (used in figure 4) is not clear.

iPSCs are generated from skin cells of different donors by reprogramming, and then differentiated into the corresponding iPSC-RPEs. hPSCs used in this study are all derived from the parental H1-iCas9 hPSC line, and then differentiated into the corresponding hPSC-RPEs. The cells in Figures 3 and 4 are all hPSC-RPEs, except for a *BEST1^WT/WT^* iPSC-RPE which is used in Figure 3a to serve as a control for the validation of *BEST1^WT/WT^* hPSC-RPE. For clarification, we have labelled the results from iPSC-RPE and hPSC-RPE cells in Figure 3a and added a paragraph in **Discussion** (Page 13).

6. The authors might consider including a key in the figure, especially where four traces are included in a single I-V plot. Although all of the information is included in the figure legend, the reader might be able to more quickly understand figure without needing to go back and forth with the figure legend when examining the data.

As the reviewer suggested, we have added labels illustrating which trace is from which experimental group in Figures 1-4 and Figure 2‒figure supplement 1.

7. The authors might also consider using labels within the figures to remind readers when the data are 1:1, 4:1 or 1:4 in figures 1-3 and where applicable. Although the text explains when the different ratios are used and the experiments are well motivated, it took us a second read to clarify these differences, which are important in the light of loss-of-function versus gain-of-function mutations.

As the reviewer suggested, we have added labels illustrating which trace is from which experimental group in Figures 1-4 and Figure 2‒figure supplement 1.